# A Method for Calculating Offsets to Ozone Depletion and Climate Impacts of Ozone-Depleting Substances

Gabrielle B. Dreyfus[1,2], Stephen A. Montzka[3], Stephen O. Andersen[1], and Richard ("Tad") Ferris[1]

[1] Institute for Governance & Sustainable Development (IGSD), Washington, DC, 20016, USA
[2] Department of Physics, Georgetown University, Washington, DC 20057, USA
[3] Global Monitoring Laboratory, National Oceanic and Atmospheric Administration (NOAA), Boulder, CO, 80305, USA

*Correspondence to*: Gabrielle B. Dreyfus (gdreyfus@igsd.org)

**Abstract.** By phasing out production and consumption of most ozone depleting substances (ODSs), the Montreal Protocol on Substances that Deplete the Ozone Layer (Montreal Protocol) has avoided consequences of increased ultraviolet (UV) radiation, and it will restore stratospheric ozone to pre-1980 conditions by mid-century, assuming compliance with the phaseout. However, several studies have documented an unexpected increase in emissions and unreported production of trichlorofluoromethane (CFC-11) and other ODSs that occurred after 2012 despite production phaseouts under the Montreal Protocol. Furthermore, because most ODSs are powerful greenhouse gases there are significant climate protection benefits in collecting and destroying the substantial quantities of historically allowed production of chemicals under the Montreal Protocol that are contained in existing equipment and products and referred to as ODS "banks". Here we present a framework for considering offsets to ozone depletion, climate forcing, and other environmental impacts arising from recent or other occurrences of unexpected emissions and unreported production of Montreal Protocol controlled substances. We also show how this methodology could be applied to the destruction of banks of controlled ODSs and GHGs, or to halon or other production allowed under a Montreal Protocol Essential Use Exemption or emergency exemption. Further, we explore a range of potential actions and roughly estimate the magnitude of offset each type of action could potentially supply for ozone depletion, climate, and other environmental impacts arising from instances of unexpected emissions or unreported production should Montreal Protocol Parties agree require remedial action.

## 1 The stratospheric ozone layer and the Montreal Protocol

The stratospheric ozone layer shields Earth against ultraviolet (UV) radiation that causes skin cancer and cataracts, suppresses the human immune system, and damages agricultural and natural ecosystems and the built environment (UNEP EEAP, 2018; Bais et al., 2018). Human-made ozone-depleting substances (ODSs) deplete stratospheric ozone, thus increasing the amount of UV radiation reaching Earth's surface. Some ODSs, primarily chlorofluorocarbons (CFCs), hydrochlorofluorocarbons (HCFCs), and halons; are also potent greenhouse gases (GHGs) (Ramanathan, 1975) (World Meteorological Organization (WMO), 2018), as are the long-lived ODS-substitutes that are hydrofluorocarbons (HFCs).

The 1987 Montreal Protocol is an international treaty that has already phased out more than 99% of the production and consumption of about 100 ozone-depleting GHGs and will soon phase down about a dozen hydrofluorocarbon (HFC) GHGs that don't contain ozone-depleting chlorine or bromine. Use of these HFCs was once thought necessary to rapidly protect the ozone layer and avoid ozone tipping points, but environmentally superior replacements are available in some applications and will soon be available in others, making their use no longer necessary in most applications. For example, the Refrigeration, Air Conditioning and Heat Pumps Technical Options Committee of the Technology and Economic Assessment Panel (TEAP) to the Montreal Protocol under the United Nations Environment Programme (UNEP) provides technical information related to alternative technologies that have been investigated and employed to make it possible to virtually eliminate use of ODS and to phasedown high global warming potential (GWP) HFCs and found in their 2022 assessment report that "[u]ltralow-, low-, and/or medium-GWP alternative refrigerants are available for all [refrigeration, air conditioning, and heat pump (RACHP)] applications and are being widely applied in some RACHP applications and regions."(TEAP, 2022a) The United States Environmental Protection Agency (EPA), has estimated that ODS phaseout under the fully revised and Amended Montreal Protocol compared with a scenario of no controls will prevent approximately 443 million cases of skin cancer, 2.3 million skin cancer deaths, and 63 million cataract cases for people in the United States born in the years 1890–2100 (EPA, 2020; Madronich et al., 2021). Global impacts are significantly higher considering that these estimates are for the US alone, representing about 4.25% of global population, and do not include the economic consequences of the full spectrum in health, agricultural productivity, and product deterioration. Even a seemingly small increase in UV radiation from unexpected emissions of unreported CFC-11 production has been estimated to contribute to an additional 31,600 to 59,800 cases of skin cancer and 170 to 340 deaths and 4,100 to 9,300 cases of cataracts that would otherwise have been avoided in the US alone (EPA, 2020). Consider also that every ecosystem would suffer adverse effects owing to any increase in damaging UV radiation (Young et al., 2021). Furthermore, the family and community consequences are far worse in societies without adequate health and where food is already in short supply (Andersen and Sarma, 2002).

In May 2018, scientists warned that emissions of CFC-11 had unexpectedly increased despite a production phaseout under the Montreal Protocol (Montzka et al., 2018). In May 2019, scientists pinpointed ~60 ± 40% of unexpected emission increase to an area in China's north-eastern provinces of Shandong and Hebei and found no evidence of a significant increase in CFC-11 emissions from any other locations where monitoring stations are sensitive to emissions on a regional scale (Rigby et al., 2019; Adcock et al., 2020). Over the course of 2018 and 2019, the unexpected emissions and unreported production of CFC-11 globally and from eastern China dropped substantially (Montzka et al., 2021; Park et al., 2021). Enhanced emissions of dichlorodifluoromethane (CFC-12) from eastern China, perhaps associated with CFC-11 production, have also been suggested (Park et al., 2021). A separate study, which analyzed 27 whole air samples collected in 2016 over Hebei province, implied new production and emissions of CFC-11, CFC-12, and 1,2-dichlorotetrafluoroethane (CFC-114) in various locations in China during spring 2016 (Benish et al., 2021). Another analysis of global atmospheric concentrations of CFC-11, CFC-12, and

1,1,2-trichloro-1,2,2-trifluoroethane (CFC-113) confirmed unexpected emissions of CFC-11, but suggested the possibility of unexpected emissions of these other gases during 2014-16 and called for further investigation of potential sources of these emissions (Lickley et al., 2021). Increases in global emission and atmospheric concentrations of several CFCs with production allowed under the Montreal Protocol for use as feedstocks in the production of hydrofluorocarbons, e.g., CFC-113a, CFC-114a and CFC-115, are also being observed, together with emission increases in CFC-13 and CFC-112a, although the driver of the increase for these latter two CFCs is unclear (Western et al., 2023).

Emissions of another Montreal Protocol-controlled substance, carbon tetrachloride ($CCl_4$), also have been substantially higher than expected after the phaseout of CFC production (SPARC, 2016). $CCl_4$ is used as a feedstock in the production of CFC-11 and CFC-12. While the ongoing $CCl_4$ emissions have not been tied to non-compliance with the Montreal Protocol, they add significantly to the ozone-depleting halogen burden of the atmosphere. Hence, stakeholders may benefit from understanding how to offset the impacts of such emissions on stratospheric ozone and its recovery.

Here we propose the idea of offsetting adverse environmental impacts arising from occurrences of unreported production of Montreal Protocol-controlled ozone-depleting chemicals and substitutes such as hydrofluorocarbons. We propose that these offsets could take the form of preventing the emission of ODS and HFCs that were legally produced and would otherwise be emitted, such as through the collecting and destroying banks of these chemicals. Other options are highlighted that could also be considered to offset the ozone depletion, climate, and other environmental impacts arising from instances of unexpected emissions or unreported production should Montreal Protocol Parties agree require remedial action. Note that an offset approach could also be applied to management and destruction of ODS banks or could be used to manage halon production allowed under a Montreal Protocol Essential Use Exemption in cases where entities are allowed to use and emit available halon banks.

An important aspect of offsetting impacts relates to the timing of the impact compared to the offset. Given the added uncertainties associated with estimating the year-to-year impacts of unexpected or illicit production and associated emission that one might hope to offset, we focus here on offsetting cumulative impacts. We note that this approach is the only possible path if actions are to be taken to offset adverse impacts that occurred in the past. We also recognize that the approach of offsetting impacts with a cumulative time-frame and not year-by-year will lead to a different time-history for an impact compared to the offset, especially when the chemical being considered for supplying an offset has a substantially different lifetime than the chemical causing the adverse impact. This latter point will likely always be true when devising an offset to an impact that has already occurred.

## 2 Usage implications on estimating the magnitude of impacts to be offset

Atmospheric observations of a long-lived gas can provide an estimate of an unexpected emission magnitude that is to be offset. For many halocarbon gases, however, anomalous emissions will represent only a fraction of the total amount of chemical produced, owing to the retention of chemical in appliances, foams, etc., as so-called "banks". Relating changes in atmospheric concentrations to production and, therefore, a more complete picture of the cumulative impact into the future as the banked chemical slowly escapes to the atmosphere, requires an understanding of how substances are produced and used. We summarize here typical historical production and uses for several ODSs for which unexpected emissions have been observed.

### 2.1 CFC-11 and CFC-12

Historically, CFC-11 was manufactured using $CCl_4$ as a feedstock and was typically co-produced with CFC-12 to optimize chemical process efficiency. One implication is that detection of unexpected CFC-11 production is likely to be also associated with unreported production of CFC-12, although large uncertainties in annual estimates of global CFC-12 emissions (4–10 Gg $yr^{-1}$) have confounded efforts to detect unusual enhancement in CFC-12 emissions in recent years (Montzka et al., 2021; Park et al., 2021). Furthermore, if there is no clandestine market for unavoidable production of either substance, the substance will likely eventually be discharged and escape to the atmosphere, since destruction would involve added cost and complication.

Prior to phaseout under the Montreal Protocol, CFC-11 was principally used as a foam blowing agent and as a low-pressure refrigerant. If used for manufacturing flexible foam, emissions of CFC-11 are immediate. If used for manufacturing rigid foam or as a refrigerant, emissions of CFC-11 are distributed over the life of the product whether in its product application or in disposal unless incinerated. The 2019 Montreal Protocol Technology and Economic Assessment Panel (TEAP) Task Force concluded that "it is likely that a resumption of newly produced CFC-11 usage in closed-cell foams in some regions was the dominant cause for the emission increase after 2012, due to technical ease and economic advantage of its use." (TEAP, 2019) The implication is that a good fraction of the emissions will lag production by many years, so the CFC-11 contained in these newly produced foams will continue to escape to the atmosphere and enhance CFC-11 emissions for many years into the future. The TEAP conclusion does not rule out increases in other historic CFC-11 uses adding to the unexpected emission increase after 2012, such as drug manufacture, uranium enrichment by gaseous diffusion, wind chambers, and other specialized experimental, analytical, and laboratory uses. The Montreal Laboratory and Analytical Use Exemption allows the continued production and import of small amounts of class I ODSs (CFCs, halons, carbon tetrachloride, methyl chloroform, methyl bromide, and bromochloromethane) (but not class II ODSs, e.g., HCFCs) for such uses as equipment calibration and biochemical research; as an extraction solvent, diluent, or carrier for chemical analysis; as inert solvent for chemical reactions; and other critical analytical and laboratory purposes (United Nations, 1994).

CFC-12 was principally used prior to phaseout as a propellant in aerosol products and as refrigerant. Propellent emissions are coincident with product use, while refrigerant emissions are small during manufacture of refrigeration and air conditioning appliances (McCulloch et al., 2003). While a larger fraction of CFC-12 refrigerant emissions are associated with leakage during installation, servicing, use, and disposal at end of the refrigeration and air conditioning appliance life than is true for CFC-11 in its main uses, there will still be significant emissions from air conditioning and refrigeration equipment long after that equipment was produced (Andersen et al., 2007).

## 2.2 CFC-113, CFC-113a, CFC-114a, and CFC-115

CFC-113 was predominantly used prior to phaseout as an electronics and aerospace solvent, with ongoing use as a feedstock. Prior to phase out, the annual emissions were roughly equivalent to production (adjusted for quantities held in inventory). While CFC-113 production has been phased out by the Montreal Protocol, CFC-113 and other ODSs, when used as feedstocks and entirely consumed, are currently exempted from calculations of controlled substances produced and consumed under the Montreal Protocol (Andersen et al., 2021). Ongoing substantial use of CFC-113 as a feedstock likely results in annual emissions in the amount of production that is not chemically converted into new chemicals. Early in the history of the Montreal Protocol, parties assumed that feedstock emissions would be *de minimis*, but they are now realized that they can be significant, arising from the manufacturing of chemicals and products such as plastics (Andersen et al., 2021). While products made with or containing CFC-11 and CFC-12 are typically easily identified, products made using CFC-113 solvent are almost impossible to identify due to evaporation of the solvent with no discernible residue.

Emissions of other CFCs allowed for production when used as feedstocks have been growing, with CFC-113a growing the fastest with 244% increase in emissions between 2010–2020, CFC-112a emissions growing by 169%, and CFC-114a growing 108% over the same period, although there are no known current uses for CFC-13 and CFC-112a (Western et al., 2023).

## 3 Quantifying ozone depletion, environmental, and climate impacts

The damages from unexpected emissions and unreported production of substances controlled under the Montreal Protocol can be quantified for the impacts related to ozone depletion for ODSs, UV radiation exposure to estimate health and environmental effects, and climate forcing for GHGs. In an ideal scenario, an offset would match the impacts year-by-year. However, this is likely to be impractical due to differences in the time-dependent impacts of different chemicals due to differences in potency and lifetimes and that unreported production and emissions of the controlled substances to be offset will likely precede any offset actions. For these reasons, we focus here on estimating cumulative impacts and offsets, although we realize that there are limitations of this approach, for example in cases where the impact is non-linearly related to the atmospheric abundance, as in the case of biological effects that depend on behavioral and other factors (Slaper et al., 1996). Specifically, we propose to use the established metrics of ozone depletion potential (ODP) and GWP when calculating offsets associated with an

emission. While the ODP is "defined as the ratio of calculated ozone column change for each mass unit of a gas emitted into the atmosphere relative to the calculated depletion for the reference gas CFC-11"(Fisher et al., 1990), it is also true that the ODP can be reliably used to estimate the cumulative impacts arising from a pulsed emission (Prather, 2002). In this way, ODP integrates the cumulative impact on the ozone column of a chemical relative to CFC-11 over the lifetime of the chemical and the timescale of secondary impacts. The ODP differs from the GWP in one important respect, however, in that the GWP reflects the ratio of a change in radiative forcing from an emission of gas relative to that same mass emission of carbon dioxide *integrated over a specific time horizon* (usually 100 or 20 years), and not over the lifetime of the chemical and its impacts. Choosing an appropriate integration period for estimating the impact and deriving an appropriate offset therefore will require a choice to be made, and this choice hinges on the relative importance of near-term vs long-term impacts. We discuss additional considerations of impacts and offset metrics in this section by type of impact and conclude with an illustrative example.

### 3.1 Ozone depletion for ODSs

The approach of offsetting, through a reduction in emission or production of an ODS, the cumulative ozone depletion arising from unexpected or illicit emissions after weighting those emissions by the ODP is supported by the near-linear relationships between cumulative emissions of a particular long-lived ODS and stratospheric ozone impacts from that ODS, both globally and over the Antarctic (Keeble et al., 2020; Fleming et al., 2020; Dhomse et al., 2019) as summarized in (WMO, 2021). This is because the impacts on stratospheric ozone of an emission roughly scale by the amount of chlorine (Cl) released in the stratosphere, all other factors (aerosol loading, etc.) being equal, so can be applied to CFC-12, CTC, and other ODS species (Dhomse et al., 2019; Keeble et al., 2020; WMO, 2021). Other metrics such as the Integrated Ozone Depletion could be used for quantifying the impact on stratospheric ozone of an emission to be offset, and use of this metric would provide results very similar to use of ODP unless the chemical being used to offset an impact had a substantially different loss frequency in the troposphere and stratosphere (Pyle et al., 2022).

### 3.2 Environmental impacts from ozone depletion

UV radiation exposure can be estimated from ozone depletion to estimate health and environmental effects. For example, the EPA used its Atmospheric and Health Effects Framework (AHEF) model to estimate that in the U.S. alone the unexpected CFC-11 emissions,[1] absent offset, would result in nearly 60,000 cancer deaths through 2100 that compliance with the Montreal

---

[1] This estimate is based on the "bank scenario", which assumes that CFC-11 emissions began increasing in 2012 above those expected under the reference WMO A1 scenario, peak around 77 Gg/yr in 2015–2017, then decline sharply through 2100. While the end date of unreported production is unclear, the report states that: "In the fourth scenario, CFC-11 emissions were estimated based on Dhomse et al. (2019), which constructs an emissions scenario curve based on initial rapid increase in CFC-11 emissions and slower release from accumulated CFC-11 banks. This scenario first takes the estimate of 13 Gg/year in new emissions due to unreported production and assumes an immediate production release rate of 15 percent followed by 3.5 percent/year. This creates a gradually decreasing emissions curve where CFC-11 emissions continue past 2100 due to releases from the accumulated bank even after production goes to zero." (EPA, 2020)

Protocol would have avoided (EPA, 2020). This and other health effects models can be extrapolated taking into account geographic location, genetic vulnerability, lifestyle differences, and access to preventative and therapeutic mitigation (Slaper et al., 1996; Longstreth et al., 1998; Struijs et al., 2010; van Dijk et al., 2013). A calculation of the health and environmental impacts from ozone depletion and global warming of emissions is beyond the scope of this paper, as the authors are unaware of simplified metrics for these impacts analogous to the metrics for estimating ozone column impacts and global warming potential.

## 3.3 Climate forcing for greenhouse gases (GHGs)

Offsets in carbon emissions are measured in tonnes of carbon dioxide-equivalent ($CO_2$-eq) using Global Warming Potentials (GWP) from the most recently published set, e.g. (Burkholder et al., 2022). The issue of timescales is also important with this metric, as the GWP involves a comparison of the cumulative climate impact over a specified time interval of a pulse emission for chemicals having different lifetimes. While 100-year GWP are most commonly used to capture the longer-term warming effects of long-lived greenhouse gases like $CO_2$ and CFCs, the use of 20-year GWP may be more relevant when considering near-term warming impacts of potent but short-lived GHGs like most HFCs. Such near-term impacts are particularly relevant to temperature goals such as limiting warming to 1.5°C with no- or limited overshoot, noting that the possibility of crossing the 1.5°C warming target of the Paris Agreement as soon as the 2030s (Abernethy and Jackson, 2022; Xu et al., 2018; Intergovernmental Panel on Climate Change, 2021). Ozone damage and enhanced UV also likely diminish the uptake of $CO_2$ by the terrestrial biosphere in its capacity as a carbon sink (Young et al., 2021). These impacts could also be included in deriving appropriate offsets if desired.

## 3.4 Illustrative offset calculation

When deriving offsets based on anomalies in emissions, it is important to remember to consider the potential for future emissions that have not yet escaped to the atmosphere (e.g., from banked chemicals that were produced illicitly but that have not yet reached the atmosphere, see earlier text). In the case of the unexpected CFC-11 emissions, the TEAP Task Force found: "the estimated cumulative total of unreported CFC-11 production is 320-700 kilotonnes in the period 2007-2019. Assuming usage in closed-cell foam production, this cumulative unreported CFC-11 production would lead to an estimated increase in the magnitude of the CFC-11 bank of 300 (266-333) kilotonnes by the end of 2019." (TEAP, 2022b) Taking the cumulative total of unreported production of 320-700 kilotonnes CFC-11, we calculate an ODP-weighted emission of 320–700 kilotonnes and $GWP_{20}$ of 2.7–6.0 $GtCO_2e$ and $GWP_{100}$ of 2.1–4.5 $GtCO_2e$ (Table 1). To calculate equivalent offsets, the formula below is considered below to derive offsets for three potential substances that are being phased out under the Montreal Protocol (Table 1).

$$\text{Mass of Chemical X (kilotonnes)} = [\text{Mass CFC-11 (kilotonnes)}] \times [\text{metric for CFC-11}] / [\text{metric for Chemical X}] \quad (1)$$

Offsetting this cumulative total CFC-11 production on ozone depletion would require preventing emission of 8,420 to 10.840 kilotonnes of HCFC-22, either through the destruction of that amount from existing banks or as reduced production allowances. In this case, the amount of HCFC required to offset the cumulative ozone impacts is greater than the amount that would be needed to offset the global warming impacts under both 20- and 100-year time horizons. For comparison, the estimated cumulative HCFC-22 production allowed under the Montreal Protocol phaseout schedule for controlled uses (excluding feedstocks) through 2040 to be on the order of 1,300 kilotonnes (Table 2). Even if recent HCFC-22 production from 2021-2023 were considered available for recovery and destruction, this would only amount to about 900 kilotonnes available for offset. Due to the low ozone depleting potential of HFCF-22, it would take the additional step of recovery and destruction of CFC banks or some other actions to offset the ozone impacts of the unexpected CFC-11 production.

Table 1. CFC-11 cumulative production in ODP and GWP and calculated mass of HCFC-22, HCFC-141b, and HCFC-142b to achieve equivalent ODP or GWP offsets. ODP and GWP values are from Table A-5 in the 2022 Quadrennial Ozone Assessment (Burkholder et al., 2022).

| CFC-11 cumulative production | | | |
|---|---|---|---|
| (kilotonnes) | ODP (kilotonnes) | $GWP_{20}$ (GtCO2e) | $GWP_{100}$ (GtCO2e) |
| 1 | 1 | 8560 | 6410 |
| 320–700 | 320–700 | 2.7–6.0 | 2.1–4.5 |
| HCFC-22 | | | |
| 1 | 0.038 | 5610 | 1910 |
| HFCF-22 offset to equal CFC-11 (kilotonnes HCFC-22) | ODP offset | $GWP_{20}$ offset | $GWP_{100}$ offset |
| | 8,420–10,840 | 488–1,070 | 1,070_2,350 |
| HCFC-141b | | | |
| 1 | 0.102 | 2590 | 808 |
| HCFC-141b offset to equal CFC-11 (kilotonnes HCFC-141b) | ODP offset | $GWP_{20}$ offset | $GWP_{100}$ offset |
| | 3,140–6,860 | 1,060–2,310 | 2,540–5,550 |
| HCFC-142b | | | |
| 1 | 0.057 | 5400 | 2190 |
| | ODP offset | $GWP_{20}$ offset | $GWP_{100}$ offset |

| HCFC-142b offset to equal CFC-11 (kilotonnes HCFC-142b) | 5,610–12,300 | 510–1,110 | 937–2,050 |
|---|---|---|---|

Table 1. CFC-11 cumulative production in ODP and GWP and calculated mass of HCFC-22, HCFC-141b, and HCFC-142b to achieve equivalent ODP or GWP offsets. ODP and GWP values are from Table A-5 in the 2022 Quadrennial Ozone Assessment (Burkholder et al., 2022).

## 4 Offsets are one option to maintain the integrity of ozone and climate protection under the Montreal Protocol

While any given instance of unexpected and unreported emissions may seem small in terms of atmospheric impacts, such impacts are cumulative and in absolute terms significant compared to other environmental violations where compensation is sought—consider for example the settlement between the US government and Volkswagen (VW) requiring VW to provide nearly US\$3 billion to an Environmental Mitigation Trust to "fully remediate the excess $NO_x$ emissions from the illegal vehicles" (Breyer, 2016). Stratospheric ozone depletion and climate-forcing offsets can compensate for unexpected and unreported production by: reducing production or emissions of an ODS produced legally prior to phaseout under the Montreal Protocol or by preventing emissions or production of an ODS not yet subject to the Montreal Protocol's phaseout requirements (e.g., $CF_3I$, $CH_2Cl_2$, or $N_2O$); and/or with respect to climate forcing, avoiding cumulative emissions or removing GHGs equivalent to the near term (20-year) forcing of the unexpected and unreported emission to help prevent triggering tipping points and longer-term climate change (Lenton et al., 2019). In the absence of chemical offsets being applied, an alternative approach might be to calculate cumulative health impacts and determine monetary compensation and/or punitive damages of loss in health, life, productivity, ecological impact, and materials degradation.

Offsets in ozone depletion are measured in tonnes of CFC-11 emission-equivalent (as an ODP-weighted emission). In the case of ozone depletion, the size of the offset estimated to be needed would ensure that the cumulative adverse impact of the unreported or illicit activity would be offset. Such an approach would contribute to ozone recovery and towards offsetting the health and environmental damage done prior to mitigation.

## 5 Potential actions that could offset the ozone depletion and climate impacts of unexpected and unreported production

In Table 2 we present a non-exhaustive list of potential actions that could be used to offset the ozone depletion, climate, and other environmental impacts arising from instances of unexpected and unreported production that Parties may agree warrant remedial action. Any of these measures could also be employed to offset the ozone and climate impacts of Essential Use Exemptions (EUEs) for ODSs other than HCFCs, and Critical Use Exemptions (CUEs) for methyl bromide, or emergency use. We provide indicative numbers on the potential available offsets for each type of action.

**Table 2. Overview of potential offset activities and indicative available offsets.**

| | |
|---|---|
| Accelerate the hydrochlorofluorocarbon (HCFC) phaseout faster than mandated by the Protocol (reducing both ozone depletion and climate forcing) | Based on baseline levels, phaseout schedule and current production in 2021 from Table 2 in MLF document 92/5 (MLF, 2023), we estimate the cumulative allowed HCFC production for controlled uses (excluding feedstocks) from 2024 through phaseout in 2040 to be on the order of 50,000 ODP tonnes, primarily HFCF-22, which is equivalent to approximately 1,300 kilotonnes of HCFC-22. Allowed production for controlled uses from 2021 to 2023 totaled about 900 kilotonnes of HCFC-22. |
| Limiting feedstock exemptions | While HCFC-22 production for controlled uses is phasing out, production for exempted feedstock uses is increasing. HCFC-22 production for feedstock uses was 56% of total reported production in 2017 (UNEP and TEAP, 2019). If feedstock production exemptions were to be revisited by the parties to the Montreal Protocol, then this sector could be considered in an offset framework.<br><br>Total annual feedstock production in 2019 was estimated at 558 ODP-weighted kilotonnes, with emissions of 15.0–18.7 ODP kilotonnes (Daniel et al., 2022). |
| Leapfrog hydrofluorocarbons (HFCs) to low-global warming potential (GWP) energy-efficient, next-generation fluids or technology (also mitigating ozone and climate forcing) | Estimated baseline annual HFC consumption for Article 5 parties for 2020–2022 was estimated to total 1,115 million tonnes $CO_2e$ using Annex F $GWP_{100}$ in Table 3-2 (UNEP and TEAP, 2023). Non-A5 parties are currently subject to a 10% reduction compared to baseline consumption levels, and in 2024 will start the 40% reduction step. For a sense of scale of potential for acceleration of phasedown, the baseline HFC consumption for the United States is 300 million metric tonnes (U.S. Environmental Protection Agency, 2023) and the European Union is 164 million metric tonnes (UNEP, 2023) (both use 2007 IPCC $GWP_{100}$ values). |
| Accelerate the HFC phasedown and transition to technologies with lower environmental impacts including non-fluorocarbon replacements (also not-in-kind -- NIK) | |

| | |
|---|---|
| Collect and destroy ODSs and HFCs banks (ozone and climate mitigation for ODS destruction; climate mitigation only for HFC destruction) | CFC-11 banks have been estimated to range from 70 to 1 475 kilotonnes for 2018, but the lower range was considered outside the range of realistic values and the higher range includes "inaccessible" banks that would be difficult to recover, such as foams in landfills (WMO, 2021), with higher estimates of 2,568 kilotonnes of CFC-11 (not including unexpected) and CFC-12 banks of 2,900 ODP-weighted kilotonnes (Lickley et al., 2020). |
| Replace inefficient air conditioners (ACs) with super-efficient, low-GWP ACs and destroy recovered ODS and HFC refrigerants | A high-end range for potential HFC offsets can be estimated from a scenario where emissions from new production and banks ceased in 2023, which would reduce cumulative emissions by 32–37 $GtCO_2e$ relative to the Kigali Amendment schedule (Liang et al., 2022). |
| Reduce production and emissions of ozone-depleting GHGs not controlled under the Montreal Protocol (i.e., $N_2O$ or $CH_2Cl_2$) or GHGs not controlled under the Montreal Protocol (i.e., $CH_4$) | Industrial emissions were 307 kt $N_2O$ (84 $GtCO_2e$) from adipic acid and 136 kt $N_2O$ (37 $GtCO_2e$) from nitric acid production in 2020; U.S. EPA estimates 80% abatement potential at break-even costs (Davidson and Winiwarter, 2023). |
| Increase the energy efficiency performance of building air conditioning and residential, commercial, and industrial refrigeration together with ODS and HFC transitions | Cumulative energy-related $CO_2$ emissions for cooling for 2023-2050 could be reduced by 47–69% through improved energy efficiency, while transitioning to low-GWP refrigerant would add a further 25–53% reduction in $CO_2e$ terms (TEAP, 2023). |

Others have implicitly suggested the usefulness of offsets: "The recovery and destruction of CFC-11 banks would not only accelerate the ozone layer recovery, but also yield climate benefits (WMO (2021). Based on the TEAP/UNEP (2019b) scenario, recovery and destruction of the active and inactive banks would reduce emissions by 1.6 Gt $CO_{2\text{-eq}}$ [100-year; 2.2 $GtCO_{2\text{-eq}}$ 20-year] between 2020 and 2060 and 2.6 Gt $CO_{2\text{-eq}}$ [100-year; by 3.6 $GtCO_{2\text{-eq}}$ 20-year] between 2020 and 2100 (see Table 5.2).

Using their estimates of much larger banks, Lickley et al. (2020) estimated that recovery and destruction of the CFC-11 and CFC-12 banks would reduce emissions by 9 Gt $CO_{2\text{-eq}}$ [100-year; 13 Gt $GtCO_{2\text{-eq}}$ 20-year] between 2020 and 2100." Recent analysis by (Lickley et al., 2022) suggests that production may have been underreported for nearly all chemicals examined, implying larger banks, and they conclude that "in terms of climate impacts, CFC-11, CFC-12, and HCFC-22 are the largest banked materials weighted by GWP100, accounting for 36 %, 14 %, and 36 % of current [ODS] banks, respectively. When

banks are weighted by ODP, CFC-11 and CFC-12 represent 46 % and halons also represent 46 % of current banked chemicals…. In terms of GWP100, CFC-11 banks largely reside in foams, whereas CFC-12 and HCFC-22 are largely in non-hermetic refrigeration. The latter may be more readily recoverable. In terms of ODP, CFC-11 foams and CFC-12 non-hermetic refrigeration remain important, along with halons which are all contained in fire extinguishers, a recoverable reservoir."[2]

## 6 Conclusion

Options are available to offset the ozone depletion and climate forcing from unexpected and unreported production and associated emissions of ozone-depleting substances (ODSs) (e.g., CFC-11, CFC-12, CFC-113, and CTC). Scientists can calculate the contribution of each option in offsetting potentially both the annual and cumulative ozone depletion and climate forcing over the atmospheric lifetime of the ODSs but we argue here that typically the most practical approach (and in some instances the only approach) will be to consider offsetting cumulative impacts without consideration of timing. The Montreal

Protocol Parties have shown creativity and flexibility in their non-compliance remedies (UNEP, 1992). Parties to the Montreal Protocol may wish to consider action on compliance to minimize ozone and climate consequences and to discourage future unexpected and unreported production.

**Author contribution**

SOA conceptualized the study. GBD, SAM, SOA, RF wrote the paper.

**Competing interests**

The authors declare that they have no conflict of interest.

**Acknowledgements**

The authors thank colleagues who helped identify the policy relevant science and the also opportunities to offset ODS and HFC emissions. We are particularly grateful to John S. Daniel, David W. Fahey, Korey G. Silverman-Roati, and Durwood

Zaelke. GBD, SOA and RF acknowledge support from the Children's Investment Fund Foundation.

---

[2] If an alternative is identified for those essential uses preserving health and safety, halons could be a recoverable reservoir .

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
