# Peer review of "A Method for Calculating Offsets to Ozone Depletion and Climate Impacts of Ozone-Depleting Substances"

_Atmospheric Chemistry and Physics, 2023_

## Author Response (AR2)

To the editors, thank you for the opportunity to resubmit as a technical note. We have revised the manuscript to align with this manuscript type with a greater focus on the approach, while also retaining the significant revisions we had made in response to the reviewer comments. See responses to those comments below from the previous version.

RC1

In this manuscript, the authors propose a framework of methods to evaluate impacts of unexpected emissions of ODS on ozone depletion and climate change. The article provides an interesting summary of methods and metrics proposed in the literature for evaluating these impacts and suggest a list of actions for offsetting them. The article is well written and documented, however I wonder if it fits in the scope of Atmospheric Chemistry and Physics journal since it is very qualitative and provide few quantitative estimates of the impact of the various proposed options for offsetting impacts of unexpected ODS emissions. In addition, the quantitative values cited in section 5 correspond to citations from the literature, e.g. WMO, 2021 or Lickley et al, 2022. The manuscript does not include any figures or tables. In order for the manuscript to better fit in the scope of ACP and be published in the journal, I suggest that the authors provide their own quantitative estimates of the various proposed options and/or their assessment of what would be the best options for the ozone depletion and for climate change issues.

Response: We thank the reviewer for the thoughtful comments and have revised the manuscript to address the issues raised. In particular we have provided quantitative examples and tables indicating the potential magnitude of offsets for the activities described. In addition we have provided additional discussion on consideration of impacts and offset metrics, including the choice to use the well-established metrics of ozone depletion potential (ODP) and global warming potential (GWP) to estimate cumulative impacts and offsets.

**Minor comments**

Page 2, line 35. The authors could elaborate on the environmentally superior replacements of HFCs

Response: We have added the following sentence and incorporated footnote 1 to elaborate on this point: "For example, the Refrigeration, Air Conditioning and Heat Pumps Technical Options Committee of the Technology and Economic Assessment Panel (TEAP) to the Montreal Protocol under the United Nations Environment Programme (UNEP) provides technical information related to alternative technologies that have been investigated and employed to make it possible to virtually eliminate use of ODS and to phasedown high global warming potential HFCs and found in their 2022 assessment report that "[u]ltralow-, low-, and/or medium-GWP alternative refrigerants are available for all [refrigeration, air conditioning, and heat pump (RACHP)] applications and are being widely applied in some RACHP applications and regions."" Citing TEAP (2022) Report of the Refrigeration, Air Conditioning and Heat Pumps Technical

Options Committee: 2022 Assessment, UNEP
https://ozone.unep.org/system/files/documents/RTOC-assessment%20-report-2022.pdf.

Page 2, line 38. Tt seems that 1890 is a typo.

Response: Thank you for the close read, however, this is not a typo. The assessment covers the population born in the United States between 1890 and 2100: "Comparing the Montreal Protocol as amended and adjusted with a scenario of no controls on ODSs showed the prevention of an estimated 443 million cases of skin cancer and 63 million cataract cases for people born in the United States between 1890 and 2100." Madronich S., Lee-Taylor J. M., Wagner M., Kyle J., Hu Z., & Landolfi R. (2021) *Estimation of Skin and Ocular Damage Avoided in the United States through Implementation of the Montreal Protocol on Substances that Deplete the Ozone Layer*, ACS Earth Space Chem. 5(8): 1876–88 https://doi.org/10.1021/acsearthspacechem.1c00183.

Page 2, line 44. cite also Young et al., 2021

Response: Thank you, we have added this citation.

Page 3, line 85. It seems that additional CFC-12 emission was not detected during the unexpected CFC-11 emission period in 2012 – 2018. Can the authors elaborate on that?

Response: We have added to section 2.1 CFC-11 and CFC-12 the highlighted phrase: "One implication is that detection of unexpected CFC-11 production is likely to be also associated with unreported production of CFC-12, although large uncertainties in annual estimates of global CFC-12 emissions (4–10 Gg yr$^{-1}$) have confounded efforts to detect unusual enhancement in CFC-12 emissions in recent years (Montzka et al., 2021; Park et al., 2021)."

Page 4, line 96. Are the mentioned experimental and analytical use controlled by the Montreal Protocol or exempted?

Response: We have added text to clarify that CFC-11 used for feedstock and process agents or in laboratory and analytical uses is not reported as production and consumption: "The Montreal Laboratory and Analytical Use Exemption allows the continued production and import of small amounts of class I ODSs (CFCs, halons, carbon tetrachloride, methyl chloroform, methyl bromide, and bromochloromethane) (but not class II ODSs, e.g., HCFCs) for such uses as equipment calibration and biochemical research; as an extraction solvent, diluent, or carrier for chemical analysis; as inert solvent for chemical reactions; and other critical analytical and laboratory purposes (Montreal Protocol Handbook, Essential Use Exemptions, Annex II)."

Page 4, line 106. The sentence starting with "Where entirely used as feedstock" is not clear. For which use is CFC-113 production exempted? The whole paragraph on CFC-113 needs to be clarified.

Response: We have revised the text to clarify that production as feedstocks and processes agents is conditionally allowed and that Class I ODSs are allowed for analytical and laboratory uses: "Prior to phase out, the annual emissions were roughly equivalent to production (adjusted for quantities held in inventory). While CFC-113 production has been phased out by the Montreal Protocol, CFC-113 and other ODSs, when used for feedstocks and entirely consumed, are currently exempted from calculations of controlled substances produced and consumed under the Montreal Protocol (Andersen et al., 2021)."

Page 5, line 143-146. We miss information for fully understand the statement. A formula could help explain on shorter time intervals the offset could be smaller or larger than the adverse impact being offset.

Response: This sentence has been removed, and we hope that the issue of time-dependencies in impact vs offset is now clearer with the revision.

Page 5, line 149-150. The authors only cite the literature. Evaluating health effects in other countries and latitudes warrants a whole new study.

Response: Agreed. We have added a sentence noting this: "A calculation of the health and environmental impacts from ozone depletion and global warming of emissions is beyond the scope of this paper, as the authors are unaware of simplified metrics for these impacts analogous to the metrics for estimating ozone column impacts and global warming potential."

Page 6, line 164. The end of the sentence is rather obscure. Global Warming Potentials are generally based on a 100 year time frame.

Response: Clarified the distinction between longer-term and near-term temperature goals in the use of 100-year GWP vs 20-year GWP: "While 100-year GWP are most commonly used to capture the longer-term warming effects of long-lived greenhouse gases like $CO_2$ and CFCs, the use of 20-year GWP may be more relevant when considering near-term warming impacts of potent but short-lived GHGs like most HFCs. Such near-term impacts are particularly relevant to temperature goals such as limiting warming to 1.5°C with no- or limited overshoot, noting that the possibility of crossing the 1.5°C warming target of the Paris Agreement as soon as the 2030s (Abernethy and Jackson, 2022; Xu et al., 2018; Intergovernmental Panel on Climate Change, 2021)."

Page 7-8, line 204 – 2017. As mentioned in the introduction of this review, a quantitative estimate of the impact of each proposed action on ozone depletion and climate change is lacking.

Response: We have made the activities into Table 2 and we have now included estimates of magnitude that helps address this point.

References:

Lickley, M. J., Daniel, J. S., Fleming, E. L., Reimann, S., and Solomon, S.: Bayesian assessment of chlorofluorocarbon (CFC),

hydrochlorofluorocarbon (HCFC) and halon banks suggest large reservoirs still present in old equipment, Atmospheric Chem. Phys., 22, 11125–11136, https://doi.org/10.5194/acp-22-11125-2022, 2022

Young, P. J., Harper, A. B., Huntingford, C., Paul, N. D., Morgenstern, O., Newman, P. A., Oman, L. D., Madronich, S., and Garcia, R. R.: The Montreal Protocol protects the terrestrial carbon sink, Nature, 596, 384–388, https://doi.org/10.1038/s41586-021-03737-3, 2021.

WMO: Report on the Unexpected Emissions of CFC-11: A Report of the Scientific Assessment Panel of the Montreal Protocol on Substances that Deplete the Ozone Layer, Geneva, Switzerland, 2021

[Figure]
 Reply
Citation: https://doi.org/10.5194/acp-2023-53-RC1
* * *
RC2

The manuscript by Dreyfus et al. discusses options to offset the ozone depletion and climate forcing impacts of additional emissions of ozone depleting substances. The manuscript rises an important and timely topic and discusses possible further action. However, it is not clear to me if the manuscript in its present form is suitable as an article in Atmos. Chem. Phys. It has more the character of a commentary, rather than a scientific research article. As a commentary, it will be a useful contribution towards the timely discussion of how to calculate impacts of ODS and options for possible offsets. As a scientific research article it does not provide enough detail and evidence for the proposed method for calculating offsets to ozone depletion and climate impacts. With the current evaluation criteria (Does the manuscript represent a substantial contribution to scientific progress: substantial new concepts, ideas, methods, or data; Are the scientific approach and applied methods valid?) I suggest to reject.

We thank the reviewer for the thoughtful comments and have revised the manuscript to address the issues raised. In particular we have provided quantitative examples and tables indicating the potential magnitude of offsets for the activities described. In addition we have provided additional discussion on consideration of impacts and offset metrics, including the choice to use the well-established metrics of ozone depletion potential (ODP) and global warming potential (GWP) to estimate cumulative impacts and offsets.

Specific comments:

I suggest to move the first paragraph(s) of Section 4 ("The idea of an offset to unexpected emissions and unreported production of ozone depleting GHGs is to collect and destroy…") to

the Introduction, to make clear from the beginning how the discussed offsets are to be understood.

Response: Agreed. Moved the opening paragraph of Section 4 to the Introduction.

There is very little discussion on the cited numbers for the resulting ozone depletion per cumulative ODS emission. How well does this linear scaling work, where/when does it break down? Are these numbers robust across models or model dependent? Are they in line with our theoretical understanding or purely empirical?

Response: The text citing those numbers has been removed, but the concept remains important to the issue of offsets. The issue of non-linearity is now mentioned as being a caveat, but would require extensive ozone modeling to address here, so is beyond the scope of the paper.

The integrated ozone depletion (IOD) diagnostic proposed by Pyle et al. (2022) may provide a more robust framework. This is briefly discussed in the present manuscript. However, it did not become clear to me, if the concept of IOD is part of the suggested method for calculating offsets or not.

Response: The IOD does supply an alternative framework for calculating offsets that wouldn't be appreciably different except for instances with very different lifetimes for the two chemicals being considered. A sentence has been added to the text to provide clarification.

It remains unclear to me, how the offset calculation takes into account ozone depletion and climate impacts at the same time. Section 3 closes with a remark on possible reduced CO2 update by the biosphere due to increased UV levels: "These impacts could also be included in deriving appropriate offsets if desired." How should this be done? How important is this effect? Are there other possible feedbacks of importance?

Response: This paper is seeking to introduce the concept of offsets, and we have added an example calculation based on the metrics that have been developed for ozone depletion and global warming. Similar metrics are not currently available for UV and environmental impacts, however, more involved modeling, such as used by EPA and by Young et al. (2021) could be used to assess these impacts for specific offset cases.

Reply
Citation: https://doi.org/10.5194/acp-2023-53-RC2

---

## Author Response (AR4)

**Referee 1**

In their revised manuscript, the authors have taken into account the previous comments. In particular, the manuscript is now considered a "Technical note"; however, I still believe that this manuscript is more of an "Opinion". Apart from that, I have only a few minor comments:

I found Table 1 not easy to understand. Please consider presenting Table 1 is an easier and clearer way with clearly defined columns.
Moreover, Table 1 currently has captions above and below.
Thank you for this comment. We have clarified the columns and rows and deleted the duplicate caption.

Line 215: brackets do not balance
Deleted extra parenthesis.

Line 346: "UNEP: Country Data, 2023." Is this a complete (and useful?) reference?
Expanded on reference to Ozone Secretariat database of Consumption of controlled substances in ODP tonnes or in $CO_2$-eq tonnes (https://ozone.unep.org/countries/data-table?report_type=0&output_type=odp-CO2e-tonnes&party%5B%5D=65&party_grouping=individual&group%5B%5D=10&period_start=1986&period_end=2022&baseline=1&group_by=group&op=GENERATE+REPORT&form_id=ozone_country_data_form__report_table_form ).

It is a bit inconsistent that WMO 2018 is included in the references, but WMO 2022 only as Burkholder et al. (2022)
Updated reference from WMO 2018 to WMO 2022.

**Referee 2**

In this resubmitted manuscript, the authors propose a framework of methods to evaluate impacts of unexpected emissions of ODS on ozone depletion and climate change. It suggests methods and metrics cited in the literature for evaluating these impacts and propose a list of actions for offsetting them. The article is well written and documented, and is publishable as a technical note since it corresponds mostly to a review of literature with suggested actions. These actions are mainly the destruction of banks, an accelerated phasedown of ODS and their HFC replacement, or the limitation of exemption use. It now includes two tables providing (1) calculation of equivalent ODS or GWP offset, and an overwiew of offset activities with an assessment of their offset.
My main criticisms and suggestions to the manuscript at this stage are the following:
- Provide equation for ODP and GWP calculations
Equations added

- Provide the context for the choice of HCFC-22, HCFC-141b and HCFC-142b in table 1, e.g. what are the main ODSs presently produced and what is the phase out schedule?
Added text explaining that the three ODS in Table 1 are those with the greatest remaining eligible production and consumption under the phaseout schedule. Added reference to TEAP

2023 supplementary report Table 4-1 with remaining eligible consumption for HCFC-141b and HCFC-142b.

- Be more specific in table 2 in the offset activities, e.g. for the HCFC-22 use as feedstock, what production would be affected by such limitation?
Table 2 provides the available information on the total annual feedstock production. It would be up to the Parties to determine the extent of reduction in emissions from feedstock production through consideration of measures, such as narrowing of feedstock exemptions. We added a reference to Andersen et al. (2021) for a discussion of narrowing feedstock exemptions, which is beyond the scope of this technical note.

For the accelerated HFC phasedown, which HFC should be targeted? Also the proposed action to reduce production of N2O, CH2Cl2 or CH4 is very vague. Why adipic acid or nitric acid productions are targeted? What would be feasible for CH4, with which impact?
A full discussion of all potential offset activities, mechanisms, and impacts is beyond the scope of this technical note, which seeks to present to concept, approach, and potential examples in Table 2.

More generally, the manuscript lacks an assessment of the most useful offsetting measures and the best options for the ozone depletion and climate change issues.
We appreciate the point and have proposed a follow-up opinion paper to the editors that would be a more appropriate forum for exploring potential offsetting measures.

Also, some references seem outdated, such as WMO (2018). The new assessment published in 2022 could be mentioned.
Updated reference to WMO 2022 and estimated remaining eligible ODS consumption from latest TEAP RTF supplementary report.

Minor comments

Page 4, line 125. Spell out acronyms
Spelled out carbon tetrachloride (CTC).

Page 4, line 126: Integrated Ozone depletion should be defined with an equation and compared to ODP.
Added equation. Noted that "Integrated Ozone Depletion (IOD) could be used for quantifying the impact on stratospheric ozone of an emission to be offset, and use of this metric would provide results very similar to use of ODP unless the chemical being used to offset an impact had a substantially different loss frequency in the troposphere and stratosphere (Pyle et al., 2022)."

Page 5, line 150: Provide examples of short-lived HFCs.
Added lifetimes of most common HFCs: HFC-134a, HFC-32, HFC-125 and lifetimes from WMO 2022.

Page 8, line 207. Provide the definition of essential use exemption and critical use exemption.

References to the specific Decisions where each of these exemptions are defined have been added. Including the full Decisions defining each exemption would significantly lengthen the text.